# Heat Treatments Effects on Nickel-Based Superalloy Inconel 713C

**Breno Boretti Galizoni** [1,]*[ ], **Antônio Augusto Couto** [2] **and Danieli Aparecida Pereira Reis** [1,3]

1 Instituto Tecnológico de Aeronáutica, São José dos Campos, CEP 12228-900, Brazil;
breno.galizoni@gmail.com
2 Instituto de Pesquisas Energéticas e Nucleares and Mackenzie, São Paulo, CEP 05508-900, Brazil;
acouto@ipen.br
3 Department of Science and Technology, Universidade Federal de São Paulo, São José dos Campos, CEP
12231-280, Brazil; danielireis@gmail.com
* Correspondence: breno.galizoni@gmail.com; Tel.: +55-19-98234-6326

**Abstract:** The purpose of this work is to study the effect of heat treatments on the microstructure of the nickel-based superalloy Inconel 713C. Three different conditions were studied and the results compared: (1) as cast; (2) solid solution treatment (1179 °C/2 h) and (3) stabilization heat treatment (1179 °C/2 h plus 926 °C/16 h). Inconel 713C is normally used in the as-cast condition, an improvement in the 980 °C stress-rupture life is often obtained by a solution heat treatment. However, the material in this condition tested under high stress at 730 °C shows a marked decreased in rupture life and ductility. The mechanical resistance in creep increases in Inconel 713C by precipitation hardening phase, with $\gamma'$ ($Ni_3Al$) formed during the heat treatments. The characterization techniques used were: chemical analysis, hardness testing, X-ray diffraction, optical microscopy and scanning electron microscopy (SEM), EDS analyzes and thermocalculation. The SEM and EDS analysis illustrated the $\gamma$, $\gamma'$ and carbides. The matrix phase ($\gamma$), has in its constitution the precipitation of the $\gamma'$ phase, in a cubic form, and in some regions, carbides were modified through the heat treatments. ($M_{23}C_6$-type) and boride ($M_3B_2$ type) identified with the use of the thermocalculation. The heat treatments increase the relative intensity of niobium in the carbides. The hardness test was not achieved because the material was overaged.

**Keywords:** heat treatment; microstructure evolution; Inconel 713C

---

## 1. Introduction

The severe crisis that hit the economy and, consequently, the aeronautics market in the early 21st century, led aircraft and engine manufacturers to develop more efficient products. New generations of aircraft had aerodynamically improved wings, increased use of composite materials and new aluminum alloys, as well as new manufacturing processes which contributed to weight reduction [1–3]. Aircraft engines provide some of the most demanding applications for structural materials. Moderns turbine engines operate at high temperatures and stresses, and engine components are often subject to damaging corrosion, oxidation, and erosion conditions. These engines convert fuel energy into propulsive thrust. During the past several decades, higher engine performance has been achieved by increasing turbine gas temperature and by increasing the efficiency of each engine stage [2,3]. Since the development of gas turbine engines for defense jet aircraft, the application of materials in high temperature has been studied. The term superalloy was first used in the mid-1940s to describe high-temperature alloys that could not only be used at elevated temperature but also maintained their strength and toughness at elevated temperature. The term referred to nickel- and cobalt-alloys [4].

Superalloys are largely employed in the manufacture of gas turbine components, such as blades, rotors, and vanes. These applications are the result of early developments, originally carried out for military and civilian aviation, which were transferred to the power generation industry. Two techniques were extremely relevant to the production and development of parts manufactured in superalloys: Vacuum furnaces technology and investment casting. Additionally, the complex geometry of gas turbine components, such as blades, and rotors, does not allow the intense use of machining processes. In this sense, the use of investment casting techniques was decisive for the success of superalloy's products with the use of Inconel 713C [5].

For most applications, Inconel alloy is specified as: Solution annealed and precipitation hardened (aged hardening). Inconel 713C is hardened by the precipitation of secondary phases (e.g., gamma prime and carbides) into the metal matrix. The precipitation of these nickel- (aluminum, titanium, and niobium) phases is induced by heat treating in the temperature range of 600 to 950 °C. For this metallurgical reaction to properly take place, the stabilizing constituents (aluminum, titanium, niobium) must be in solution (dissolved in the matrix); if they are precipitated as some other phase or are combined in some other form, they will not precipitate correctly and the full strength of the alloy with not be reach realized. To perform this function, the material must first be solution heat treated (solution annealed is a synonymous term) [6–8].

Binczyka and Śleziona [9] studied the cooling rate in hardness measures of Inconel 713C. The analysis of the microhardness of the alloying constituents yielded the expected results, where an increased cooling rate raises the macrohardness of samples. The phase of the lowest hardness was γ phase, followed by γ′ phase. The highest hardness had the eutectic carbides. So, more carbides, the macrohardness increases [10–23].

The objective of this work is to evaluate the effect of heat treatment in the microstructure of Inconel 713C, through chemical analysis, hardness test, X-ray diffraction, optical microscope and scanning electron microscopy (SEM), EDS analyzes and thermocalculation.

## 2. Material and Methods

### 2.1. Material

The material used was the nickel-based superalloy Inconel 713C supplied by Açotécnica S.A. as an investment casted one-inch diameter rod. The chemical composition is (in wt.%) 73.0Ni-0.14C-13.0Cr-4.8Mo-2.3Cb(Nb)-0.75Ti-6.0Al-0.010B-0.10Zr in % weight. [10]. The casting of the alloy is performed in vacuum melt furnace. The minimum tensile strength and yield strength at 0.2% offset are 758 MPa and 689 MPa respectively [23].

### 2.2. Heat Treatment

Two heat treatments were realized, one solution treatment performed by heating with 10 °C per min rate until 1176 °C for 2 h followed by air cooling. The second treatment, stabilization heat treatment, was realized in samples that were already solid solution heat treated and it consists of heating with 10 °C per min rate until 927 °C for 16 h followed by air cooling. The heat treatment routes are shown in Figure 1.

The heat treatments were performed in a tubular furnace Lindberg/Blue M (Thermo Scientific, Waltham, MA, USA) (100 V/50 A/50 kW). After the heat treatments, three conditions of Inconel 713C were obtained: as cast, solid solution heat treatment and stabilization heat treatment.

### 2.3. Chemical Analysis

The materials in three conditions were tested, and the results were compared with the international specification of the alloy, AMS 5391 [23]. The analyses of optical emission spectroscopy were carried out in Spectromaxx equipment (Spectro, Kleve, Germany).

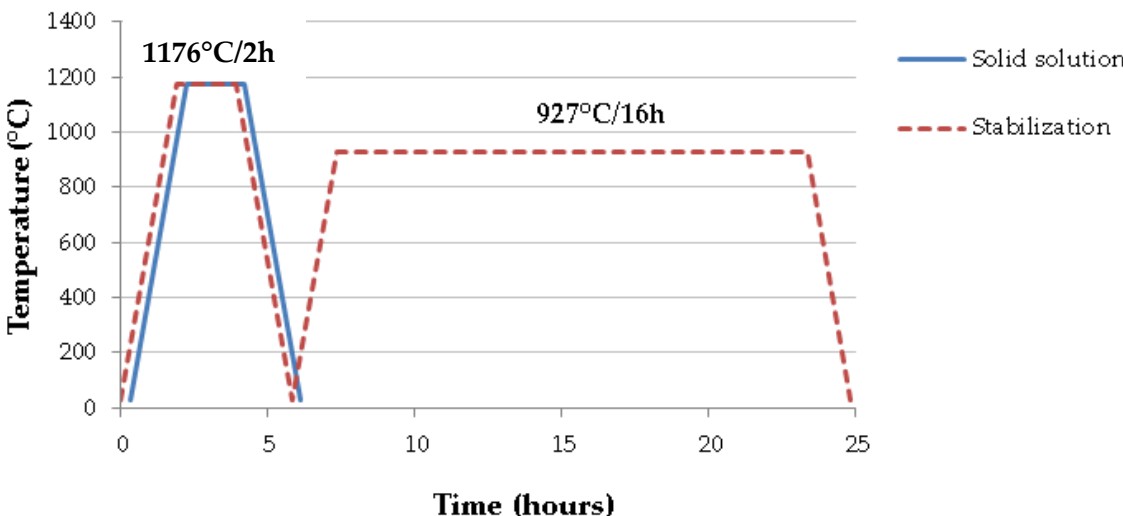

**Figure 1.** Heat treatments curve realized in the Inconel 713C samples.

*2.4. Hardness Test*

Rockwell C hardness measurements (HRC) (Future-Tech Corp, Kawasaki, Japan), with a 15N test load, were performed using Tech LC 200RB equipment. The measurements were obtained on the three samples of each conditions, as-cast, solid solution heat treatment and stabilization heat treatment.

*2.5. X-Ray Diffraction*

The X-ray diffraction experiments at room temperature were performed in Panalytical Empyrean equipment (Malvern Panalytical, Royston, UK) using Ni-filtered Cu-Kα radiation, angular interval (2θ) from 10° to 120°, angular step of 0.02°, and counting time of 30 s.

*2.6. Microstructural Analysis*

The analyses of the Inconel 713C microstructure after the heat treatments were carried out in the cross sections of the specimens by optical microscopy. The samples were cut, mechanically polished and electrochemically etched—3% sulfuric acid at 3 V for 5–10 s. A Zeiss Axio Imager Scope A1 microscope (Carl Zeiss, Munich, Germany) was used for microstructure visualization. The scanning electron microscope (SEM) (Tescan, Kohoutovice, Czech Republic) coupled with a Tescan Vega 3 XMU with Energy-dispersive X-ray spectroscopy (EDS) (Tescan, Kohoutovice, Czech Republic).

*2.7. Thermocalculation*

The phase stability evaluation was carried out using Thermo-Calc Software (Database Version 8, Thermo-Calc Software, Solna, Sweden) and TTN18/Ni-based superalloys, database version 8 [24]. The purposes of the calculation was predicting the phase transformation temperatures, the elemental composition of the microstructures in off-equilibrium conditions (ScheilModel).

## 3. Results and Discussion

*3.1. Chemical Analysis Results*

The characterizations by chemical analysis of the main elements, the percentage by weight, were performed in three samples, each with a different heat treatment—As cast, solid solution heat treatment and stabilization heat treatment. The results obtained (in wt.%), as well as the comparison with the international standard AMS 5391 [23], are indicated in Table 1.

**Table 1.** Results of the chemical analyze of the Inconel 713C samples compared to the international standard AMS 5391.

| Chemical Analysis of Inconel 713C | | | | |
|---|---|---|---|---|
| Elements (wt.%) | AMS 5391 | As Cast | Solid Solution | Stabilization |
| Cr | 12.00–14.00 | 13.32 | 13.58 | 13.70 |
| Mo | 2.80–5.20 | 3.85 | 3.93 | 3.96 |
| Al | 5.50–6.50 | 5.62 | 5.72 | 5.60 |
| Ti | 0.50–1.00 | 0.67 | 0.70 | 0.67 |
| C | 0.08–0.20 | 0.11 | 0.11 | 0.11 |
| B | 0.005–0.015 | 0.009 | 0.01 | 0.009 |
| Zr | 0.05–0.15 | 0.076 | 0.085 | 0.083 |
| Si | max. 0.50 | 0.15 | 0.15 | 0.13 |
| Mn | max. 0.25 | 0.012 | 0.012 | 0.007 |
| Fe | max. 2.50 | 0.89 | 0.86 | 0.89 |
| Cu | max. 0.50 | 0.004 | 0.005 | 0.004 |
| Ni | balance | balance | balance | balance |

The comparison of the results obtained in the chemical analyzes with the AMS 5391 standard met the requirements described in the standard. The chemical analysis of the material in the as-cast condition considered to be satisfactory is essential to evaluate the quality of the material manufactured by Açotecnica S.A. and to enable the continuity of the work.

The formation of matrix phase $\gamma$ (CFC—austenitic) occurs by the presence of chemical elements nickel, chromium, and molybdenum. On the other hand, the presence of the aluminum and titanium elements are responsible for the formation of intermetallic hardening phases of type $A_3B-Ni_3(Al,Ti)$, the best known being the $\gamma'$ phase. The precipitation of the $\gamma'$ phase in the matrix phase $\gamma$ contributes to the increase of the mechanical resistance; this is due to the resistance of the precipitates to the movement of the dislocations present in the material.

### 3.2. Hardness Measurements

Table 2 shows the average hardness data of five samples in each of the three heat treatment conditions (as cast, solid solution heat treated and stabilization heat treatment).

**Table 2.** Results of the Rockwell C hardness tests in the samples.

| Inconel 713C | Hardness (HRC) |
|---|---|
| As cast | $36 \pm 1$ |
| Solid solution | $40 \pm 1$ |
| Stabilization | $37 \pm 1$ |

The alloy Inconel 713C is a casted alloy, and does not have a hardness specified in international standards (AMS 5391), however a hardness range of 30–42 HRC (Rockwell C) is defined [23]. All results obtained in the hardness tests in the three heat treatment conditions were within this range.

For the samples heat-treated, a variation in the hardness was evidenced. The solid solution heat treated samples showed an increase of less than 10% in the hardness values obtained. On the other hand, the stabilization heat treated samples showed a decrease in the hardness in comparison with the solid solution heat treated, returning to the level of hardness of the as casted samples, like overaged process.

### 3.3. X-Ray Diffraction Results

The X-ray diffraction analysis was performed in the three heat treatment conditions, and the results showed in all conditions peaks at the same crystallographic angles, but with different intensities.

The peaks were found at 2θ = 44.5°, 51.9°, 76.4°, 92.9° and 98.5°, and there are the same found in data-base of γ-phase with different lattice parameter as shown in Figure 2.

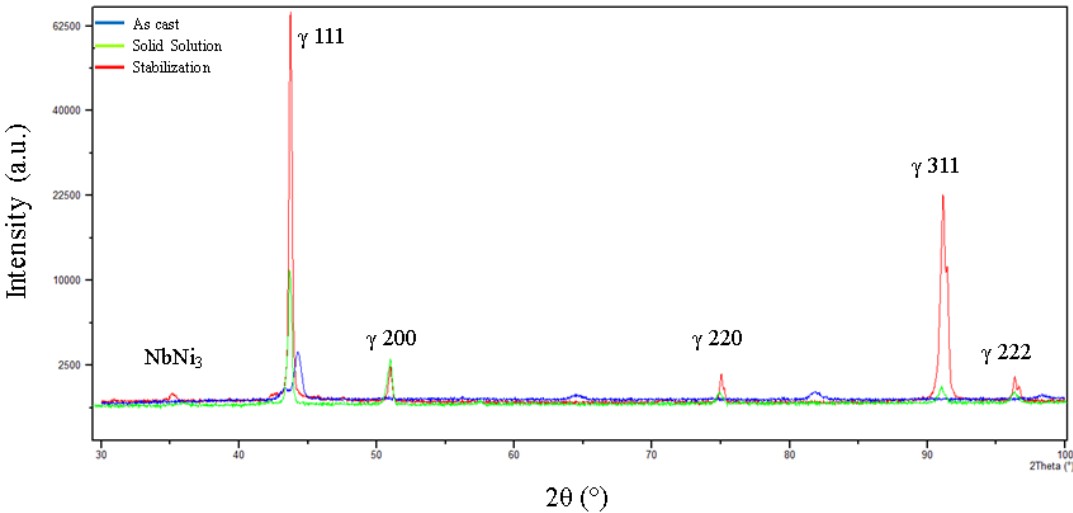

**Figure 2.** X-ray diffraction images of the samples (as cast, solid solution heat treated and stabilization heat treated).

As such, the stabilization heat treated condition foments the γ-phase with the lattice parameters [111] and [311]. The solid solution heat treated condition foments the same lattice parameters, but with less intensity. The small peak at 35.5° found in the stabilization heat treated condition is the carbide $NbNi_3$.

### 3.4. Microstructural Analysis

The typical microstructure of Inconel cast products is a γ-phase matrix whose dendritic structure is highly influenced by the casting process, such as cooling and solidification rate.

The presence of defects such as porosity, segregation, and inclusions of non-metallic is normal in materials in the as-cast state, without thermomechanical treatments. In optical microscopy analysis, it is possible to observe the porosity between dendrites, as well as segregations and carbides.

In Figure 3, it is possible to observe the elongation of the dendrites as a consequence of the casting process, but there is no difference in the microstructure after the heat treatments.

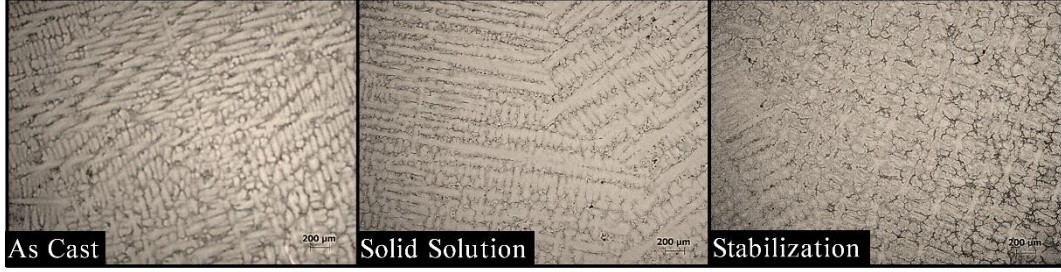

**Figure 3.** Images obtained by optical microscopy of the Inconel 713C material as cast, solid solution heat treated and stabilization heat treated condition.

The microstructure with interdendritic precipitation of carbides (light phase) is highlighted in Figure 4.

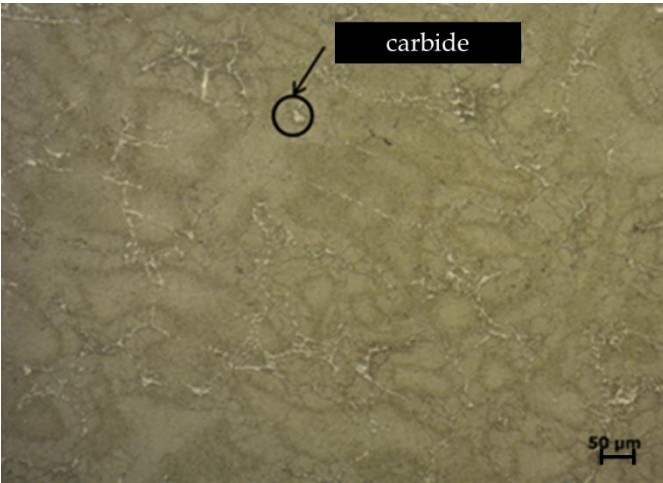

**Figure 4.** Carbides identified in the image obtained from optical microscopy in the stabilization heat treated condition.

Scanning electron microscopy (SEM) was performed on samples in the three heat treatment conditions. The samples were chemical etched, and the objective of the analysis was to observe microstructural differences occurring with the heat treatments. Micrographs were obtained using secondary electrons.

Figure 5 shows the sample in the as cast condition, where it is possible to observe the matrix phase $\gamma$, the darkest region, and the $\gamma'$ phase, the clearest region. The $\gamma'$ phase is present in the intragranular form.

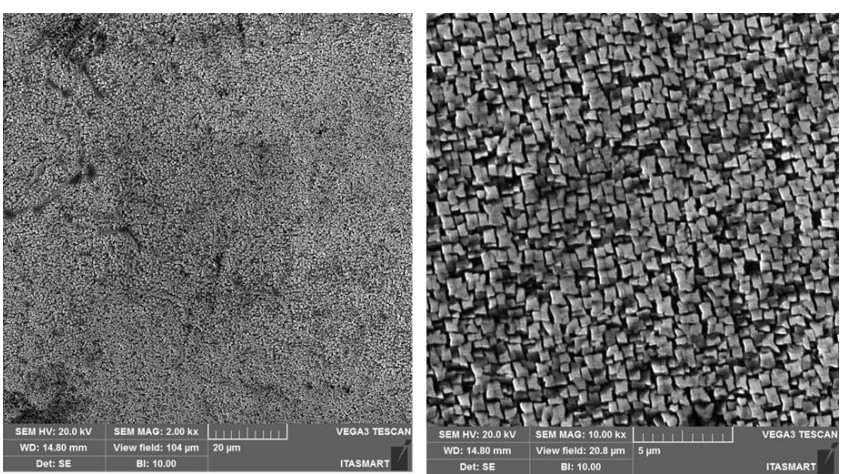

**Figure 5.** SEM of the as cast condition.

With the subsequent heat treatments, it was possible to show the higher distribution of the $\gamma'$ along the matrix phase $\gamma$, as can be observed in Figure 6. This distribution is related to the mobility of these atoms, leaving the concentrated $\gamma$ matrix phase to the $\gamma'$ mainly in the stabilization heat treated condition.

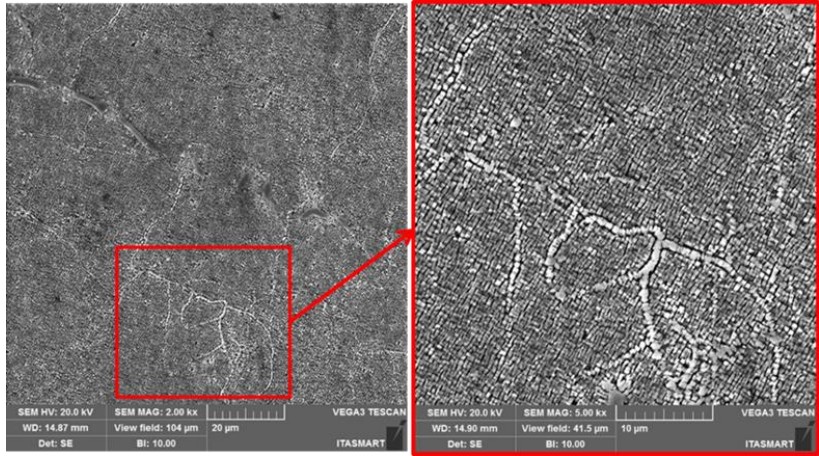

**Figure 6.** SEM of the stabilization heat treated condition.

### 3.5. Energy-Dispersive X-Ray Spectroscopy (EDS)

Linear EDS analysis was performed to confirm the variation of the chemical elements in the microstructure, the results are described in Figure 7. The analysis presents the variation of the chemical elements along a line passing through the fields of the matrix phase, along with the dispersed phase and the carbide veins.

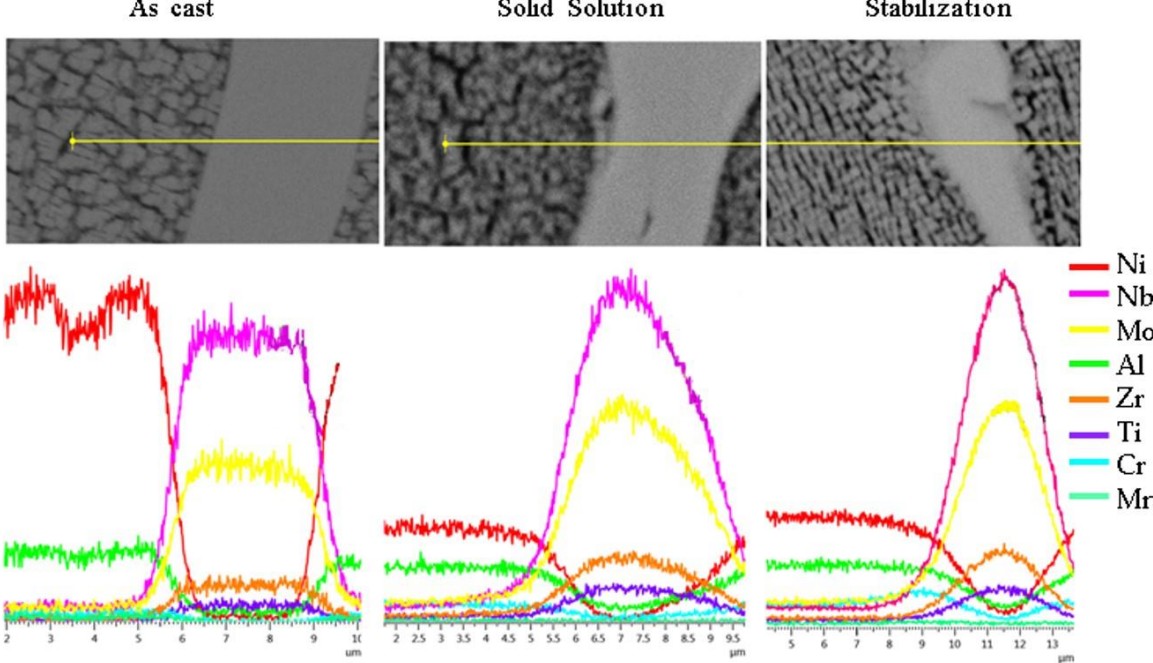

**Figure 7.** EDS of the samples in the three heat treatment conditions.

Figure 7 shows that in the γ and γ′ phase there is a much higher presence of nickel than in the carbides, as well as aluminum, elements responsible for the formation of both phases. In the carbides, there is a considerable presence of niobium, molybdenum, zirconium, and titanium, these being the elements present in the carbides MC, $M_{23}C_6$ and $M_6C$.

With the heat treatments, the niobium concentration reduced in the matrix phase. With the treatment of stabilization heat treated, there is a higher concentration of niobium in the carbides, increasing its concentration in relation to the others, such as molybdenum, zirconium, and titanium.

### 3.6. Thermocalculation of Phase Stability

The simulation results of the percentage weight of each phase as a function of temperature are shown in Figures 8 and 9. Figure 8 shows the major phases and the Figure 9, the minority phases. The liquidus and solidus temperatures are 1350 and 1280 °C, respectively, indicating a solidification range of 70 °C. The first phase to solidify from the liquid is the matrix phase γ, the first carbides, MC-type, started to segregate around 1360 °C, with the ending of the liquid phase solidification, both phases coexist at the end of solidification, forming a microstructure γ plus MC-type carbides in this range of temperature. The Ti, Nb, and Cr present in the alloy are carbide-forming elements.

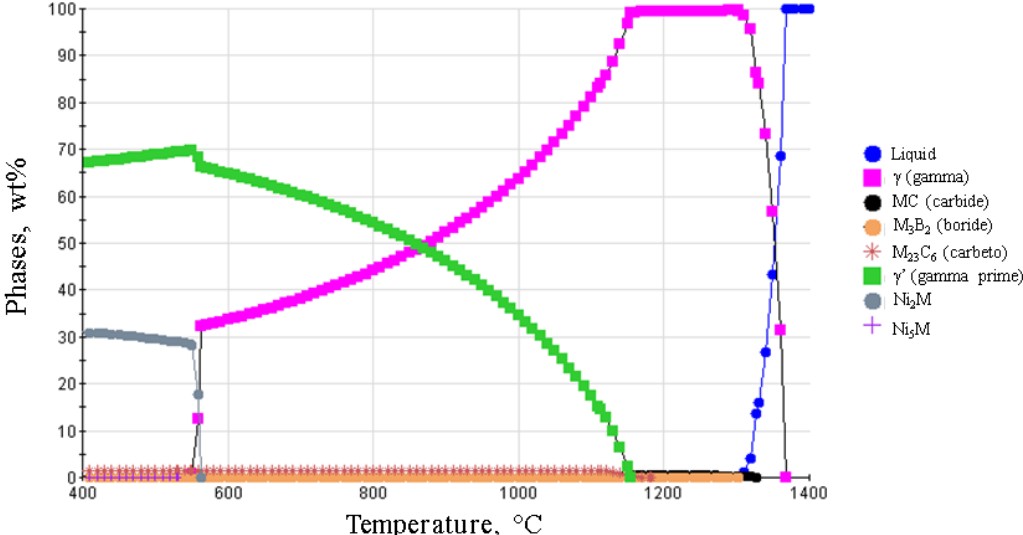

**Figure 8.** Percentage (in wt.%) of the majority phases vs temperature in Inconel 713C.

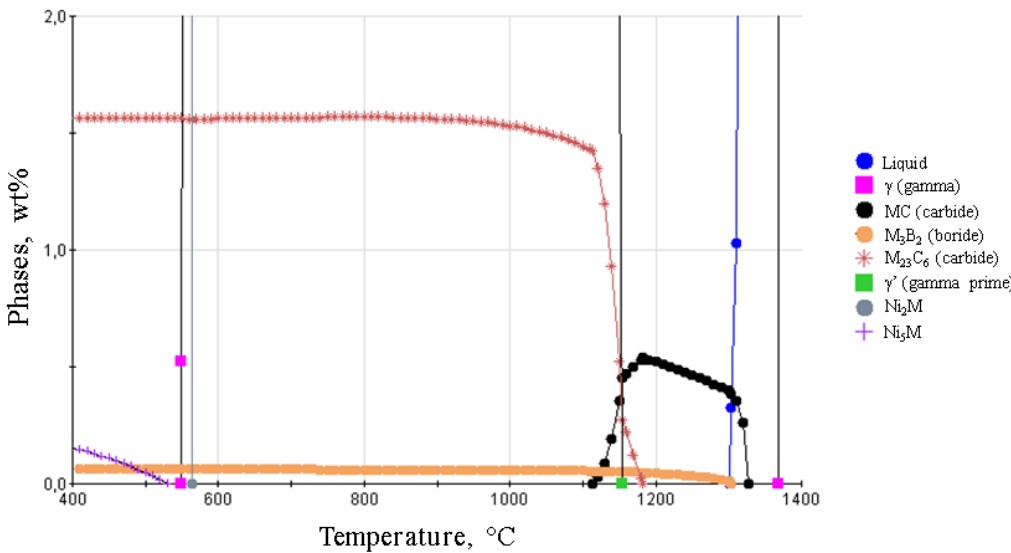

**Figure 9.** Percentage (in wt.%) of the minority phases vs temperature in Inconel 713C.

The predicted γ' solvus temperature is around 1175 °C where the phase starts increasing with the decrease in the γ phase percentage. The minority phase graphic (Figure 9), shows the stability of the carbides, the MC-type is stable from 1360 to 1160 °C, at temperatures lower than 1200 °C the $M_{23}C_6$ is more stable, and a small percentage of boride $M_3B_2$-type is also stable. All of these phases are modelling by the software and should be confirmed using Transmission Electron Microscope (TEM).

## 4. Conclusions

It can be concluded that the heat treatments of solution and stabilization heat treatment of the Inconel 713C alloy alter the microstructure of the alloy with the formation of carbides rich in NbNi₃, presented in the XRD, 35.5° (2θ) at the stabilization heat treated condition. There is a change in the niobium concentration of the carbides, these hardening phases of the alloy with the heat treatments. The heat treatments increase the relative intensity of niobium in the carbides, presented by the EDS. The final microstructure expected is matrix phase γ, solubilized γ' with a small percentage of carbide ($M_{23}C_6$-type) and boride ($M_3B_2$ type), which is shown in the SEM images and identified with the use of the thermocalculation of the phase stability.

**Author Contributions:** Conceptualization: B.B.G., A.A.C. and D.A.P.R.; Validation: B.B.G., A.A.C. and D.A.P.R.; Formal Analysis: B.B.G., A.A.C. and D.A.P.R.; Investigation: B.B.G., A.A.C. and D.A.P.R.; Writing-Review and Editing: B.B.G. and D.A.P.R.; Supervision: D.A.P.R.

**Funding:** This research received no external funding.

**Acknowledgments:** Thanks for FAPESP, CAPES and CNPq.

**Conflicts of Interest:** The authors declare no conflict of interest.

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
