# Peer review of "Heat Treatments Effects on Nickel-Based Superalloy Inconel 713C"

_metals, doi:10.3390/met9010047_

Reviewer 1 Report

The authors have investigated the effect of heat treatments on the microstructure and properties of Inconel 713C, specifically as cast, solution treatment and a stabilising treatment. Overall, the article is well put together, there are some points and queries that must be addressed before publication as given below:

The paper requires a thorough check of grammar, tense and punctuation.

Line 17 – phase, such with

Line 18 – used was were: chemical analysis, hardness testing, X-ray diffraction, optical microscopy

Line 22 – veins of carbides were modified with through the heat treatments

Line 29 – aluminium alloys

Introduction – this section requires work; the first 3 paragraphs should be concisely confined into one paragraph.

Line 55 – Can you provide the dimensions of the as-received casting/rod?

Line 57 – The casting of the alloy is a procedure performed

Line 60 – How were the samples cooled/what were the cooling rates?

Line 61 – Confusion as to whether it is being termed a solution treatment or a solubilization treatment & a stabilization or aging treatment. Needs to be consistent throughout article.

Line 66 – Figure 1 – time axis could have a proper scale/unit and use fonts/labelling consistent with rest of article

Line 75 – How many hardness tests were performed to ensure reliable/consistent results?

Line 87 – coupled with Energy-dispersive X-ray spectroscopy (EDS) coupled.

Line 91 – calculation were was

Line 102 – a comment on the decrease in Mn content in the stabilized/aged variant should be given

Line 124 – why do the hardness values return to the level of the casted samples?

Line 129 – Figure 2 – graph axis labels need to be much clearer, and the spectrum clarity should be improved

Figure 3/4/5/6/7 – Increase mag bar scales for clarity in each figure

Line 149 – The samples were embedded and attached – unsure what is meant by this?

Line 167 – the results were are

Line 171 – Figure 7 We shows

Figure 8 – need to be able to see all alloy element colours – one legend for all 3 samples using the same colour scheme would be best.

Line 174 – can you identify these different carbide formations in the microstructure? MC, M23C6 and M6C?

Line 181 – Fig.s Figs. 8 and 9

Figure 8/9 – remove titles as given in caption description

Line 200 – The main objective of these heat treatments is the improvement of the mechanical properties of the alloy studied. – has this objective been achieved, the small amount of hardness tests are the only indication of mechanical properties presented.

Author Response

The authors have investigated the effect of heat treatments on the microstructure and properties of Inconel 713C, specifically as cast, solution treatment and a stabilising treatment. Overall, the article is well put together, there are some points and queries that must be addressed before publication as given below:

The paper requires a thorough check of grammar, tense and punctuation.

Line 17 – phase, such with

ok

Line 18 – used was were: chemical analysis, hardness testing, X-ray diffraction, optical microscopy

ok

Line 22 – veins of carbides were modified with through the heat treatments

ok

Line 29 – aluminium alloys

ok

Introduction – this section requires work; the first 3 paragraphs should be concisely confined into one paragraph.

ok

Line 55 – Can you provide the dimensions of the as-received casting/rod?

ok

Line 57 – The casting of the alloy is a procedure performed

ok

Line 60 – How were the samples cooled/what were the cooling rates? Air cooling (added to the papper)

Line 61 – Confusion as to whether it is being termed a solution treatment or a solubilization treatment & a stabilization or aging treatment. Needs to be consistent throughout article.

Ok - Stabilized

Line 66 – Figure 1 – time axis could have a proper scale/unit and use fonts/labelling consistent with rest of article

ok

Line 75 – How many hardness tests were performed to ensure reliable/consistent results?

Ok – 3 each conditions

Line 87 – coupled with Energy-dispersive X-ray spectroscopy (EDS) coupled.

ok

Line 91 – calculation were was

ok

Line 102 – a comment on the decrease in Mn content in the stabilized/aged variant should be given

Ok – the decrease of Mn was not so big to matter

Line 124 – why do the hardness values return to the level of the casted samples?

Ok – because of na overaged

Line 129 – Figure 2 – graph axis labels need to be much clearer, and the spectrum clarity should be improved

ok

Figure 3/4/5/6/7 – Increase mag bar scales for clarity in each figure

ok

Line 149 – The samples were embedded and attached – unsure what is meant by this?

ok

Line 167 – the results were are

ok

Line 171 – Figure 7 We shows

ok

Figure 8 – need to be able to see all alloy element colours – one legend for all 3 samples using the same colour scheme would be best.

ok

Line 174 – can you identify these different carbide formations in the microstructure? MC, M23C6 and M6C?

No, Transmission Electron microscope could help (next step)

Line 181 – Fig.s Figs. 8 and 9

 ok

Figure 8/9 – remove titles as given in caption description

ok

Line 200 – The main objective of these heat treatments is the improvement of the mechanical properties of the alloy studied. – has this objective been achieved, the small amount of hardness tests are the only indication of mechanical properties presented.

Ok, the conclusion was improved

Reviewer 2 Report

Dear Editor in regard to the manuscript at hand I would like to make the following comments as examples

1- The introduction is very general and not specific to the material in this work or the processes involved. So from the introduction, it is not quite clear if there has been any previous work and if there has what differs this study from them. The authors basically use 4 paragraphs of general superalloys introduction and then they are at reference number 25! with no mention of the alloy, heat treatment of Inconel 713C and the properties/microstructures from previous studies!

2- There are minor language inaccuracies and less that desired choice of words which makes following the manuscript rather difficult. There are however some sentences that are difficult to comprehend as well.

below are a few examples:

- solubilization treatment should be solution treatment

- line 102-104 not clear what intended to mean

-line 149 the samples were embedded and attacked?!

3- the data is not sufficiently described, captions are vague and too short.

Figure 2: The X-ray diffraction data should be resolved for the phases present and indicate what is the gamma phase gamma prime, carbides and/or tcp phases. The data without description of the phases is useless. The say the peaks are the same almost at the same position but the intensity is different! Some of them are but if the intensity is not the same what is the reason! Minimal information on this! And some of the peaks close to 2theta=49° are not the same.

Figure 3: The micrographs belong to different samples as I understand them and depending on the region and the cut (sample cross section) the image of the dendritic/interdendritic microstructure can be different and therefore the conclusion in lines 140 and 141 are difficult to follow. Perhaps if the same region of the same sample had gone through the solution treatment and aging the conclusion would be more acceptable.

Figures 6 and 7: they both show the same thing and those cubes are not carbides but most likely gamma prime phase. It seems they have made a big mistake in this regard. 

4-There are basic issues this work in my opinion, some of which I point out in the following,

Lines: 105-107. How could the chemical composition be varied by heat treatment in the bulk!  Has there been cases where the bulk chemical composition changed after heat treatment? As there is no mention of the location of the samples plus considering the nature of the chemical analysis one can conclude it has been done over multiple D and ID regions and for a relatively large sample. Therefore, the above-mentioned sentences become redundant at best.

5- the connection between the results and the conclusions is not clear. there is mention of carbides of two types but how were these two phases identified and evaluated.

Author Response

Dear Editor in regard to the manuscript at hand I would like to make the following comments as examples

1-      The introduction is very general and not specific to the material in this work or the processes involved. So from the introduction, it is not quite clear if there has been any previous work and if there has what differs this study from them. The authors basically use 4 paragraphs of general superalloys introduction and then they are at reference number 25! with no mention of the alloy, heat treatment of Inconel 713C and the properties/microstructures from previous studies!

Ok, the introduction was improved, with heat treatment of Inconel 713C and literature results

2- There are minor language inaccuracies and less that desired choice of words which makes following the manuscript rather difficult. There are however some sentences that are difficult to comprehend as well.

below are a few examples:

- solubilization treatment should be solution treatment

Ok, solution and stabilized treatment

- line 102-104 not clear what intended to mean

Ok, corrected

-line 149 the samples were embedded and attacked?!

Ok, corrected

3- the data is not sufficiently described, captions are vague and too short.

Figure 2: The X-ray diffraction data should be resolved for the phases present and indicate what is the gamma phase gamma prime, carbides and/or tcp phases. The data without description of the phases is useless. The say the peaks are the same almost at the same position but the intensity is different! Some of them are but if the intensity is not the same what is the reason! Minimal information on this! And some of the peaks close to 2theta=49° are not the same.

Ok, improved

Figure 3: The micrographs belong to different samples as I understand them and depending on the region and the cut (sample cross section) the image of the dendritic/interdendritic microstructure can be different and therefore the conclusion in lines 140 and 141 are difficult to follow. Perhaps if the same region of the same sample had gone through the solution treatment and aging the conclusion would be more acceptable.

Ok, corrected

Figures 6 and 7: they both show the same thing and those cubes are not carbides but most likely gamma prime phase. It seems they have made a big mistake in this regard.

Ok, corrected

4-There are basic issues this work in my opinion, some of which I point out in the following,

Lines: 105-107. How could the chemical composition be varied by heat treatment in the bulk!  Has there been cases where the bulk chemical composition changed after heat treatment? As there is no mention of the location of the samples plus considering the nature of the chemical analysis one can conclude it has been done over multiple D and ID regions and for a relatively large sample. Therefore, the above-mentioned sentences become redundant at best.

Ok, corrected

5- the connection between the results and the conclusions is not clear. there is mention of carbides of two types but how were these two phases identified and evaluated.

Ok, corrected

Reviewer 3 Report

Abstract: The abstract is disorienting: it seems that the heat treatment tested is a novelty and no references are quoted. IN713 material finds application both in the as-cast (even in the directional state) and after HT. The proposed thermal treatment (is a two step SOL+STA treatment) are already known and referenced in a rich literature.

 line 25 a typing error in: Introduction.

lines 31-36:  second paragraph: why there are 9 references for this simple sentence? please select the more appropriate and insert the right references in the right position.

line 61 - 62: no cooling rate is reported. Probably is air-cooling as reported in many literatures, isn't?

line 62: the heat treatment step at 927°C is often reported as a "stabilizing step" and the ageing is a further step at lower temperatures (800°C...)

line 71: You state: "The received materials were tested…" but in line 56 you state: " the material used was…….a conventionally casted rod. "  Revise as applicable.

Table 1: why you were expecting that the Thermal treatment should have an effect on the alloy chemical composition. This is a useless activity: seems more a quality check that some companies request for the HT certification.

line 61: no cooling rate is reported.

lines 109-113: You discuss the precipitation of several phases but no peak is indicated in the X-ray diffration spectra, fig.2.

Fig.4: without an EDS analysis report or RX the carbide identification should be recalled in the text and discussed deeply, Probably you need a TEM approach for the phase' identification.

lines 105 - 107: delete.

line 196 201: the complex phases identification Probably is derived form the thermodynamic calculations, please quot ethe software used and specify that are modelled phases and not proven by TEM analysis.

You paper do not prove that there you find a new Heat Tretment with some improvment. The work is very similar to a quality check. 

The title "Heat Treatment effects on...IN713".. should reflect the content and results: where are the effects on corrosion, tensile, fatigue, creep properties?

exemplum of missing REF:

1. Pratt and Whitney Aircraft Company, Ma terials Engineering Section.
2. "Haynes Alloy N o. 713C , " D ata Sheet F -30; 154A, Marc h 1960.
3. A. E. C ers, A. A. Bl at herwick, " Fatigue and St res s-Rupture Propertie s of Inc onel 713C , V-57C " WADD Technical Report 60-426, July 1960.

Author Response

Abstract: The abstract is disorienting: it seems that the heat treatment tested is a novelty and no references are quoted. IN713 material finds application both in the as-cast (even in the directional state) and after HT. The proposed thermal treatment (is a two step SOL+STA treatment) are already known and referenced in a rich literature.

Improved the abstract

 line 25 a typing error in: Introduction. ok

lines 31-36:  second paragraph: why there are 9 references for this simple sentence? please select the more appropriate and insert the right references in the right position.

All references were improved

line 61 - 62: no cooling rate is reported. Probably is air-cooling as reported in many literatures, isn't?

The cooling rate was reported

line 62: the heat treatment step at 927°C is often reported as a "stabilizing step" and the ageing is a further step at lower temperatures (800°C...)

Change „aged“ for „stabiliyzed“

line 71: You state: "The received materials were tested…" but in line 56 you state: " the material used was…….a conventionally casted rod. "  Revise as applicable.

Revised

Table 1: why you were expecting that the Thermal treatment should have an effect on the alloy chemical composition. This is a useless activity: seems more a quality check that some companies request for the HT certification.

No change of chemical composition was expected

line 61: no cooling rate is reported.

The cooling rate was reported

lines 109-113: You discuss the precipitation of several phases but no peak is indicated in the X-ray diffration spectra, fig.2.

X-Ray spectra was improved

Fig.4: without an EDS analysis report or RX the carbide identification should be recalled in the text and discussed deeply, Probably you need a TEM approach for the phase' identification.

TEM will be the next step

lines 105 - 107: delete.

Ok

line 196 201: the complex phases identification Probably is derived form the thermodynamic calculations, please quot ethe software used and specify that are modelled phases and not proven by TEM analysis.

Done

You paper do not prove that there you find a new Heat Tretment with some improvment. The work is very similar to a quality check.

The title "Heat Treatment effects on...IN713".. should reflect the content and results: where are the effects on corrosion, tensile, fatigue, creep properties?

The next step is the creep properties study

exemplum of missing REF:

1. Pratt and Whitney Aircraft Company, Ma terials Engineering Section. ok

2. "Haynes Alloy N o. 713C , " D ata Sheet F -30; 154A, Marc h 1960.

3. A. E. C ers, A. A. Bl at herwick, " Fatigue and St res s-Rupture Propertie s of Inc onel 713C , V-57C " WADD Technical Report 60-426, July 1960.

Reviewer 4 Report

1. A few literature references about heat treatment of nickel-based superalloys and in particular of Inconel 713C for the improvement of the mechanical properties, should be included in the Introduction. The Introduction only makes a few general remarks about the use of superalloys in aeroengines. The authors should also summarize the results and findings of previous literature studies and their relevance in the work reported at the present paper.

2. Divisions in the x-axis of the plot in Figure 1 are missing.

3. Peak identification should be performed on the XRD patterns shown in Figure 2. The authors should also explain the shift if any in the peak position between the several samples studied. The authors mention that the patterns of all samples exhibit the same peaks (but at different angles) but while the red and green patterns show peaks and also strong ones at the angles of ~52°, ~75°, ~92° and ~96° the blue pattern does not show any peaks at those angles. The blue pattern shows peaks at ~64° and ~83° but the red and green patterns do not show any peaks at those angles. The red pattern shows a peak at ~35° but the other two patterns do not show any peak at that angle. Explanation about those differences from pattern to pattern should be given in the paper.

4. The authors state in lines 175-176 that with the heat treatment there is a reduction in the niobium concentration in the carbides but the graphs of Figure 7 show that the relative intensity of the niobium signal in the carbides increases instead, in the heat treated samples.

5. The conclusions extracted from the phases percentage versus temperature graphs of Figure 8 and 9 should be linked and discussed with the conclusions extracted previously in the paper from the optical, SEM images and EDS results.

6. The organization of the abstract of the paper should be improved. The conclusion that the hardness of the alloy after the solubilized treatment becomes higher should also be mentioned in the abstract.

Author Response

1. A few literature references about heat treatment of nickel-based superalloys and in particular of Inconel 713C for the improvement of the mechanical properties, should be included in the Introduction. The Introduction only makes a few general remarks about the use of superalloys in aeroengines. The authors should also summarize the results and findings of previous literature studies and their relevance in the work reported at the present paper.

Literature about heat treatment and resultson hardness of Inconel 713C were added.

2. Divisions in the x-axis of the plot in Figure 1 are missing.

Done

3. Peak identification should be performed on the XRD patterns shown in Figure 2. The authors should also explain the shift if any in the peak position between the several samples studied. The authors mention that the patterns of all samples exhibit the same peaks (but at different angles) but while the red and green patterns show peaks and also strong ones at the angles of ~52°, ~75°, ~92° and ~96° the blue pattern does not show any peaks at those angles. The blue pattern shows peaks at ~64° and ~83° but the red and green patterns do not show any peaks at those angles. The red pattern shows a peak at ~35° but the other two patterns do not show any peak at that angle. Explanation about those differences from pattern to pattern should be given in the paper.

Done

4. The authors state in lines 175-176 that with the heat treatment there is a reduction in the niobium concentration in the carbides but the graphs of Figure 7 show that the relative intensity of the niobium signal in the carbides increases instead, in the heat treated samples.

Done

5. The conclusions extracted from the phases percentage versus temperature graphs of Figure 8 and 9 should be linked and discussed with the conclusions extracted previously in the paper from the optical, SEM images and EDS results.

Done

6. The organization of the abstract of the paper should be improved. The conclusion that the hardness of the alloy after the solubilized treatment becomes higher should also be mentioned in the abstract.

Done

Round  2

Reviewer 3 Report

Dear Authors,line 12th: still a ageing treatment appers.

Why you have insert the Research keyword:  "creep" ? The paper does not deal about creep.

Probably "microstructure evolution" and "thermal treatments" should be more appropriate.

The paper has been really improved with a more clear method and quality of results presentation.

The paper is interesting for the description of what happens to the 713 microstructure during the different thermal state. It is very good material for a lecture. 

Nevetheless, my personal doubt is coming from your last sentence of the "Conclusion Chapter" where you state that: "The main objective of these heat treatments is the improvement of the mechanical properties of the alloy studied, with  the hardness test it was not achieved because an overaged of the material."

At this point if nothing of "positive" has been proved to publish….I would suggest to the Authors to re-present the paper after having optimise the thermal teatment cycle. But I know that it is a really hard topic.

Author Response

Why you have insert the Research keyword:  "creep" ? The paper does not deal about creep.

Next step will be creep tests

Probably "microstructure evolution" and "thermal treatments" should be more appropriate.

Ok

The paper has been really improved with a more clear method and quality of results presentation.

Ok

The paper is interesting for the description of what happens to the 713 microstructure during the different thermal state. It is very good material for a lecture.

Ok

Nevetheless, my personal doubt is coming from your last sentence of the "Conclusion Chapter" where you state that: "The main objective of these heat treatments is the improvement of the mechanical properties of the alloy studied, with  the hardness test it was not achieved because an overaged of the material."

The sentence was corrected

At this point if nothing of "positive" has been proved to publish….I would suggest to the Authors to re-present the paper after having optimise the thermal teatment cycle. But I know that it is a really hard topic.

Reviewer 4 Report

The authors revised their paper accordingly so now the paper is ready to be published.

Author Response

The authors revised their paper accordingly so now the paper is ready to be published.

OK

Round  3

Reviewer 3 Report

Lines 12, 193 and 213 still recall the "aging treatment". In other sections of the paper the same thermal treatment is correctly recalled as stabilizing… please, armonize.

Author Response

Lines 12, 193 and 213 still recall the "aging treatment". In other sections of the paper the same thermal treatment is correctly recalled as stabilizing… please, armonize.

Corrected